

**What if the 25th October 2011 event that stroke Cinque Terre**
**(Liguria) had happened in Genova, Italy? Flooding scenarios,**
**hazard mapping and damages estimation.**
**Francesco Silvestro[1*], Nicola Rebora[1], Lauro Rossi[1], Daniele Dolia[1], Simone Gabellani[1],**
**Flavio Pignone[1], Eva Trasforini[1], Roberto Rudari[1], Silvia De Angeli[1,2], Cristiano**
**Masciulli[3]**
[1]{ CIMA research foundation, Savona, Italy}
[2] {WRR Programme, UME School, IUSS-Pavia, Italy}
[3] {IREN, Genova, Italy}
Corresponding author: Francesco Silvestro
mail: francesco.silvestro@cimafoundation.org
CIMA Research Foundation (www.cimafoundation.org)
University Campus, Armando Magliotto, 2. 17100, Savona, Italy
Tel. +39 019230271,     fax. +39 01923027240
**Abstract**
During the autumn of 2011 two catastrophic very intense rainfall events affected two different
parts of the Liguria Region of Italy causing various flash floods. The first occurred in October
and the second at the beginning of November. They became two "school cases" studied by
many scientists around the world and they awaken the interest of the local authorities and of
the civil protection actors regarding these type of calamities. Due to the large amount of
damages and the numerous victims, they caused a general increase of the sensibleness of the
citizens of the stricken areas regarding the natural hazards.





Two main considerations were done in order to set up this work. The first consideration is that
various studies demonstrated that the two events had a similar genesis and similar triggering
elements. The second very evident and coarse concern is that two main elements are needed
to have a flash flood: a very intense and localized rainfall event and a catchment (or a group
of catchments) to be affected. Starting from these assumptions we did the exercise of mixing
the two flash floods ingredients by putting the rainfall field of the first event on the main
catchment stroke by the second event that has its mouth in correspondence of the biggest city
of the Liguria Region: Genova. A complete framework was set up to quantitatively carry out a
"what if" experiment with the aim of evaluating the possible damages associated to this event.
The approachcombines a probabilistic rainfall downscaling model, a hydrological model, a
2D hydraulic model and a proper methodology for damages estimation. This leads to the
estimation of the potential economic losses and of the risk level for the people that stays in the
affected area.
The results are interesting, surprising and in such a way worrying: a rare but not impossible
event (it occurred about 50km away from Genoa) would have caused huge damages estimated
between 120 and 230 million of euros for the affected part of the city of Genova, Italy and
more than 17000 potentially affected people.
Key words: flash floods, hazard, extreme rainfall, damage estimation, risk, urban hydrology.
**1   Introduction**
Flash floods are one of the most disastrous natural hazards that affect citizens in many part of
the world causing high risk for them and for their goods and activities. Many types of flash
floods exist but in a great number of cases they are caused by very intense (i.e. 50-150 mm/h)



and localized rainfall events that persist on the same area for hours (i.e. 4-12 hrs) causing
large accumulation of precipitation and fast response of small catchments (O(Area) $10^0$ to $10^3$
km$^2$) (Quevauviller, 2014). Many authors focused on the analysis of these events, their
genesis and their ground effects (Amengual et al, 2007; Barthlott and Kirshbaum, 2013;
Gaume et al., 2009; Marchi et al., 2009; Delrieu et al., 2006; Massacanad et al., 1998; Roth et
al., 1996), and lot of research was carried out to improve their predictability in terms of
rainfall with Numerical Weather Prediction Systems (NWPSs) (Buzzi et al., 2013; Fiori et al.,
2014) and in terms of streamflow (Alfieri et al., 2012; Siccardi et al., 2005; Silvestro and
Rebora, 2014; Versini et al., 2014) even referring to hydrological nowcasting techniques
(Borga et. al, 2011; Liechti et al., 2013; Silvestro et al., 2015a)
During the autumn 2011 two flash floods stroke the Liguria Region of Italy causing a total of
19 victims and a large amount of damages. The first flash flood occurred the 25th October
2011; it affected the Cinque Terre coastal towns of Monterosso and Vernazza on the Eastern
Liguria Region and caused the flooding of Magra river.  The second event occurred 9 days
later, the 4[th] November, at about 50 km of distance and mainly affected the city of Genova
with the flooding of Bisagno creek (see Figure 1).
Figure 1
These two disastrous events were widely studied during the last five years, among the others
Silvestro et al. (2012) provided an hydrological description of the 4[th] November event
highlighting the efficacy of the forecast approach adopted by the local authorities, Rebora et
al. (2013) gave a detailed analysis of what happened based on a wide collection of observed
data, Buzzi et al. (2013) conducted a series of experiments based on a Numerical Weather
Prediction System (NWPS) to understand the genesis of the two events, Nardi and Rinaldi





(2014) analyzed the changes in space and time of channel patterns in response to the major
flood of the Magra basin during the 25[th] October event, Davolio et al. (2015) analysed the
improvements in the flood forecast of the two events due to the horizontal spatial resolution
increasing of a Numerical Weather Prediction System (NWPS) used to trigger a probabilistic
flood forecast chain.
Some of the authors of this work were involved as co-authors in many of the aforementioned
manuscripts and recently a very simple but interesting question arose: what would have been
the impacts if the storm event of 25[th] October had hit the city of Genova?
This is a reasonable question, in fact various authors (Buzzi et al., 2013; Rebora et al., 2013;
Fiori et al., 2014) demonstrated that the two events had similar characteristics and a similar
genesis, in addition many of the conditions that triggered the rainfall event were the same.
We tried to answer to this question by setting up a complete flood forecasting chain that
combine a rainfall downscaling model, a hydrological model, a 2D hydraulic model and a
methodology to estimate damages.
The rainfall downscaling model and the hydrological model are part of the flood forecasting
framework presented in Silvestro et al. (2015b) and already employed to study the
predictability of a flash flood event. The rainfall field observed on the 25[th] October 2011 on
the east part of Liguria is artificially moved on the Bisagno creek, and after aggregation at
different spatial and temporal scale is downscaled to generate possible streamflow scenarios
that affect the Genova city; this is to the knowledge of the authors, a quite novel way to set up
a "what if" experiment, in fact, on one side it allows to use a real event (not built with
standard methods based on the generation on synthetic events), on the other side it allows to
account for the uncertainties and possible variability of spatial and temporal patterns at small





scales (i.e. 1-8 km, 10-60 min) of a rainfall field with a certain volume of precipitation and a
certain spatial-temporal structure at medium and large scales (i.e. 8-30 km, 60-360 min)
In order to produce a damage assessment analysis, a sub-set of the streamflow scenarios are
used as input to a 2D hydraulic model to estimate the related hazard maps and then, using
information about exposure, an appropriate  methodology is applied to estimate the potential
damage and the risk level for the population. This latter is based on a standard approach but a
series of novel elements was introduced in order to adapt the method to the particular study
area.
Currently the planning and designing of structures and infrastructures which have the purpose
of mitigating the flood risk is carried out based on the estimation of peak flow with a certain
return period T (as an example in Italy the reference T is 200 yrs), but no indications on the
evolution of the discharge event are provided. Given a return period, different assumptions
concerning the evolution, duration of the event (shape of hydrograph, total volume, etc) can
make a real difference in terms of impacts. The presented work demonstrates that quantitative
indications on possible direct impacts can be obtained, at least in some cases, following a
"worst case" scenario perspective based on real possible events. The presented approach is
robust and it faces the problem in a probabilistic way giving possible flooding scenarios
starting from a real precipitation event.
In this way a multi-disciplinary approach was implemented in order to answer to the initial
scientific question that is: what if the 25[th] October 2011 event that stroke Cinque Terre
(Liguria) had happened in Genova, Italy?
The paper is organized as follows: section 2 describes the study area and the hydro-
meteorological data set, section 3 shows the material and models used to carry out the





experiments while in section 4 the results are reported, and finally the paper concludes in
section 5 with the discussion and conclusions.
**2    Hydro-meteorological data set and study area**
Bisagno Creek is placed in the center of the Liguria Region in northern Italy (Figure 1),
it drains a total area of approximately 98 km$^2$ and it is characterized by steep slopes due
to the a mountainous topology given its proximity to the Apennines. The minimum and
maximum elevations are 0 and 1100 m respectively, while the mean elevation is about
370 m. The majority of the Bisagno basin is covered with vegetation characterized by
forest, meadows and brushes, but the last 10 kilometres of its riverbed are heavily
urbanized; there are residential areas, factories and infrastructures which are exposed to
a high risk of flooding. Along the last 1.5 kilometres, towards the mouth, the river flows
under a cover.
The territory of Liguria is monitored by a meteorological network, named OMIRL –
"Osservatorio Meteo-Idrologico della Regione Liguria". It is the official network
managed by the Civil Protection Agency of Liguria Region and it is part of the Italian
raingauge network managed by the Italian Civil Protection Department (Molini et al,
2009). This system provides rain gauge measurements with 5-10 minutes timesteps. The
network counts a total number of about 200 instruments over the region reaching an
average density of 1 rain gauge/40 km$^2$. Stations with other sensors (temperature,
radiation, wind, air humidity, etc.) are present, even though their densities are lower
than the rain gauges density.
Bisagno Creek is a very well instrumented/monitored catchment with a rain gauge
density of about 1 rain gauge/10 km$^2$.



For the analyzed basin, level gauge data are available at the cross section Passerella
Firpo, that has an upstream area of about 93 km$^2$. The level data is combined together
with a rating curve in order to estimate the observed streamflow.
The Liguria Region (Figure 1) is covered by a Doppler polarimetric C-band radar,
located on Mount Settepani at an altitude of 1386 m, that works operationally with 10
minutes scansion time (e.g. time interval when radar data are available). Rainfall fields
are provided with 1x1 km spatial resolution.
## 3  Material and models
### 3.1  Flood Forecast Framework
The Flood Forecast Framework (hereafter FFF) is described in Silvestro et al. (2015b), and it
is made by two elements: i) RainFARM (Rebora et al. 2006a, 2006b) which is a rainfall
downscaling model used for generating an ensemble of precipitation fields that are consistent
with large scale predictions issued by meteorological models (Laiolo et al., 2013) and/or by
expert forecasters (Silvestro et al. 2011); ii) *Continuum* (Silvestro et al. 2013; Silvestro et al.,
2015c) which is a continuous distributed hydrological model.
The setting and the parameters of the *Continuum* model are obtained from previous
application (Silvestro et al., 2015b). The spatial resolution is 90 m and the temporal resolution
is 10 minutes. The considered reference model section correspond to the location of Passerella
Firpo level gauge, here the drainage area is 93 km$^2$. The model is run using meteorological
observation from ground stations starting from 1th January 2011 in order to estimate the
values of the state variables at the beginning of the event.


We supposed to know the total volume of precipitation at a certain large scale (Rainfall
Volume: RV) deriving it by the observations.
The rainfall field occurred during the 25$^{th}$ October 2011 was estimated by the radar rainfall
estimation merged with rain gauge data using the Conditional Merging (CM) technique
described in Sincalir and Pegram (2005), This rainfall field is named "true rainfall field"
(TRF). The algorithm is applied at hourly scale. The TRF is artificially moved in order to
affect the Bisagno basin with the following approach: the point where the accumulated
rainfall over 24 hours has the maximum value was made coinciding with the centroid of the
basin (see Figure 2).
Figure 2
The RainFARM parameters are estimated directly by the radar rainfall fields in order to
determine the correct spatial and temporal characteristics of the rainfall event.
A domain DV of 32 x 32 km centered where the accumulated rainfall over 24 hours has the
maximum value, was considered for computational reasons.
In order to account for the spatial-temporal variability and uncertainty of the rainfall the TRF
is aggregated (obtaining the field AF) on the DV at different time and spatial scales RS (from
fine to coarse scales) so that the total volume of rainfall of AF is conserved and equal to the
volume of TRF.
The spatial and temporal aggregation scales are chosen in order to account for the possible
uncertainties related to the temporal and spatial distribution of the rainfall and to easily
compute Fast Fourier Transform (FFT) (Rebora et al., 2006a):
Spatial Scales (km): 1, 2, 4, 8
Temporal Scales (min.): 10, 30, 60, 180, 360



The RVs are then disaggregated with RainFARM producing N equi-probable rainfall
scenarios at the radar time and spatial resolution (1 km, 10 minutes) that are used to generate
N equi-probable streamflow scenarios by the Continuum model (N=500).
For the sake of clarity we report the scheme of FFF in Figure 3
We can state that the analysis is mainly made by the following steps:
1. Aggregation of TRF on DV at fixed time and spatial scales (RS) obtaining AF
2. Downscaling AF on radar spatial and temporal resolution with RainFARM obtaining

8       N equi-probable rainfall scenarios

3. Using the N equi-probable rainfall scenarios as input to Conitnuum to produce N equi-

10       probable streamflow scenarios

Figure 3
**3.2   Hydraulic model: TELEMAC-MASCARET**
TELEMAC-MASCARET (http://www.opentelemac.org/) is an integrated suite of solvers for
applications in the field of hydraulic modelling. It is managed by a consortium of core
organizations. The suite contains different modules and in this work Telemac-2D is used. It
solves the shallow water equations, also known as the Saint Venant equations, using the
finite-element or finite-volume method and a computation mesh of triangular elements. It can
perform simulations in transient and permanent conditions. This software has many fields of
application and is widely used for both research and technical purposes. In the maritime
sphere, particular mention may be made of the sizing of port structures, the study of the
effects of building submersible dikes or dredging, the impact of waste discharged from a
coastal outfall or the study of thermal plumes. In river applications, mention may also be



made of studies relating to the impact of construction works (bridges, weirs, and tubes), dam
breaks, flooding and transport of decaying or non-decaying tracers.
**3.3    Damage estimation**
Damage computations was carried out through the RASOR (Rapid Analysis and
Spatialization Of Risk) platform (Rudari 2015, Koudogbo et al. 2014), which enables
multi-hazard risk analysis for full cycle disaster management. RASOR integrates
diverse data and products across hazards, it allows to easily update exposure data and to
make scenario-based predictions to support both short and long-term risk-related
decisions.
A conventional damage model, based on stage(m)-damage(%) vulnerability curves was
implemented to compute building damage related to each flood scenario. Damage
assessment considers physical and economic damage at structures and their content.
Besides physical and economic damage, an estimation of the population potentially
involved in the area was also given. A simple downscaling methodology was
implemented to obtain population distribution at building scale in areas with different
hazard levels.
3.3.1   Exposure-building
Very detailed exposure data were obtained merging institutional information with EO-
based and crowd-sourced geographic information and virtual surveys. Buildings were
classified according to their occupancy class (usage), as required by the vulnerability
model (see vulnerability paragraph below).
Official information from real estate registry and census (year 2011) were updated
through high-resolution optical imagery and cross-compared with crowd-sourced



dataset such as Open Street Map (http://www.openstreetmap.org). Inconsistencies found
in the comparison of the two datasets were fixed thanks to field and virtual surveys.
Moreover, from real estate registry and census datasets it is impossible to distinguish
mixed occupancy buildings. In fact, it is very common the case of buildings with
commercial activities (like shops, stores, banks, etc...) at the ground floor and dwelling
at upper floors. In the same way, no information was provided on the presence of
basement. While this type of information might play a minor role for other hazards, in
case of flood it is relevant as it changes the response of the building in terms of damage.
In this case, field and virtual surveys were realized to recognize these features and
classify them in new building classes. The whole process led to an accurate description
of the assets in the areas affected by the flood. The original occupancy classes by
HAZUS-MH database distributed from FEMA (US Federal Emergency Management
Agency) were extended as shown in Table 1.
Table 1

## 3.3.2 Exposure-population

Quantifying population exposure as a step for conducting spatially-explicit risk assessment
requires to map the spatial distribution of population with adequate spatial-temporal
resolution. Since natural hazards can affect urban areas in a very selective manner, only fine-
scale population data can provide an accurate estimate of the affected population (Deichmann
et al., 2011). Data on resident population (census tracts or global population data sources such
as WorldPop - http://www.worldpop.org.uk/, Gridded Population of the World, and Global
Rural-Urban Mapping Project by NASA, LandScan by UT-Battelle and United States
Department of Energy) are not normally available at building scale. Moreover, due to its



dynamic nature, the estimation of people presence in each building is quite complicated as it
is affected by many variables, such as hour of the day, level of productivity in the area, main
traffic patterns, etc.
In literature several methodologies are proposed to downscale population to fine scales, some
examples are: choropleth method, areal interpolation method, dasymetric method, and
statistical approach for population distribution in urban area (Bhaduri, et al., 2007; Holt et al.,
2004; Langford et al., 2008; Wu et al., 2005, S. Freire 2010).
In this study, a top-down approach is employed to spatially disaggregate and distribute the
population from official census and statistics for nighttime and daytime periods, by adapting
the methodology proposed by S. Freire and C. Aubrecht (2012).
Population is split into three classes: night-time population (equal to the residential
population); daytime residential population; and daytime worker and student population.
Total daytime population distribution results from the sum of the daytime population in their
places of work or study and the population that remains at home during the day. The latter is
obtained by multiplying the night-time distribution by the ratio of resident population who,
according to official statistics by the National Statistics Institute (ISTAT, 2011), does not
commute to work or school. Daytime population is then distributed into buildings, which are
considered the main aggregation places; a buffer around the building is considered to take into
account also of people which could be in the proximity of the building. Daytime residential
population is then equally distributed among residential building storeys while daytime
commuting workers and students are distributed into non-residential building storeys.





### 3.3.3  Vulnerability-building
A classical damage model, based on stage(m)-damage(%) vulnerability curves was
implemented to compute losses associated to each flood scenario. HAZUS-MH database
provides one of the most complete collections of stage-damage curves. Water depth-damage
functions in the HAZUS library are separately provided for structure (load-bearing systems,
architectural, mechanical and electrical components, and building finishes) and for content.
Different curves are available for different occupancy classes.
Starting from this collection, several curves were added to take into account additional classes
such as mixed occupancy (e.g. retail trade and residential) and presence of basement (see
Table 1). In order to create curves for mixed occupancy and multiple storeys residential
occupancy classes the following procedure was applied. The first part (from 0 to 3m) of the
residential curve for one-floor building (RES1) from HAZUS is intended to be representative
of each floor of a generic multi-story residential building. Under the assumption that each of
the N floors represents, in percentage of damage terms, 1/N of the total building damage, for
the construction of an N-story residential building it is necessary to sum this curve N times,
taking care to weigh each addend by multiplying by 1/N. The same hypothesis and the same
procedure apply to mixed-type buildings with commercial activities at the first floor (retail
trade or restaurant, etc.) and apartments on the other floors: in this case, for the first floor, the
first part of the curves for commercial building is used (e.g. COM1, COM8, etc.), while for
each of the other floors the residential part of RES1 is summed (N-1) times. In this case
different weights for different occupancy types can be used, as in general the value for
commercial floors is bigger than the one for residential floors.





Figure 4 shows a comparison between three water depth – damage curves for content: retail
trade (COM1) building [blue], mixed retail trade (COM1) at first floor & RES at second floor
[red], mixed retail trade (COM1) at first floor & residential (RES) at second and third floor
[green].
Figure 4
The new set of curves covers all the possible types of buildings in the flooded area.
Physical damage obtained by application of stage–damage functions can be
transformed into economic losses (ED) using replacement cost per square meter.
$$ED[€] = PD * A * RC * (n + b) \qquad (1)$$
where:
$PD\ [\%]$    is the physical damage
$A\ [m^2]$    is the area of the building footprint
$RC\ \left[\frac{€}{m^2}\right]$    is the replacement cost per square meter
$n$    is the number of floors

$$b = \begin{cases} 0 & \textit{if the building has not a basement} \\ 1 & \textit{if the building has a basement} \end{cases}$$

Two different lumped replacement costs are assigned for structure damage and content
damage: 500€/m2 for structure replacement costs, and 400€/m2 for content
replacement costs. Those costs were derived considering typical damage caused by
flood (replacement of floor, doors and window fixtures, sewage and electric systems,
finishes, plaster, etc.) and the local market prices indicated by the regional authority
(Unioncamere, 2014).


### 3.3.4 Vulnerability-population

Despite the enormous impacts of floods, there is relatively limited insight into the factors that determine the loss of life caused by flood events. In the literature several methods have been developed to assess the loss of lives due to flood events and to identify mitigation measures (DeKay ML, McClelland, 1993; Jonkman et al., 2008). In general these methods consist of a quantitative relationship between the flood characteristics (such as water depth, velocity) and the mortality in the flooded area.

In order to compare possible impacts on population for different scenarios, four hazard zones (very high, high, moderate, low flood hazard) based the human instability in floodwaters. In fact, practical experiments (Abt et al., 1989; Karvonen et al., 2000) show that in flow conditions $0.5 < v < 3$ m/s and $0.3 < h < 1.5$ m the average human instability threshold in floodwaters corresponds to $hv = 1.35$ $m^2/s$, (Jonkman et al., 2008). This is the threshold that differentiates the "high flood hazard" vs "moderate flood hazard" zones. A further thresholds (upper and lower) identify two other classes: "very high flood hazard" (very high where water level and velocity) and "low flood hazard" (low water level and velocity). The resulting four flood hazard zones can be ranked as follows:

*Very high hazard zone: if $hv \geq 5$ $m^2/s$ and $v \geq 2$ m/s*

*High hazard zone: if $h \geq 0.2$ m and $hv > 1.35$ $m^2/s$*

*Moderate hazard zone: if ($0 < h < 0.2$ m and $hv > 1.35$ $m^2/s$) or ($h \geq 0.2$ m and $h < 0.5$ m and $v > 1$ and $hv < 1.35$ $m^2/s$) or ($h > 0.5$ m and $hv < 1.35$ $m^2/s$)*


*Low hazard zone: if (h > 0 m and h<0.2m and hv < 1.35 m²/s) or (h >0.2m and h <0.5m and*
*v < 1m/s):*
For each zone potentially affected, population is computed taking into account where
the population is located during the day and the night at building level (see Exposure
paragraph). This method can give useful indications especially in relative terms when
comparing different scenarios.
**4   Results**
**4.1   FFF**
The results are shown using box plot representation. Figure 5 shows the box plot of the
500 peak flows generated with FFF compared with the mean peak flow of the sample of
500 realizations represented by the blue diamonds. Each panel refers to a different
spatial RS (RSs), while on the x-axis the temporal RS (RSt) is reported (the case with
RSs=1 km and RSt=10 minutes is obviously not considered since it corresponds to the
resolution of the original field).
Figure 5
It is noticeable the fact that the $Q_p$ varies from 1200 to 1800 m³/s considering the 25% and
75% percentile of the box especially for spatial aggregations RSs 1 and 2 km, while the
mean $Q_p$ is between 1400 and 1600 m³/s. This means that the considered rainfall field
could lead to a peak flow with a return period T larger than 200 yrs, Q(T=200 yrs)≅ 1300
m³/s  (Boni et al., 2007; Provincial Authority of Genoa, 2001). Just to have some terms of
comparison: the 4[th] November 2011 flood led to a peak flow around 750-800 m³/s


(Silvestro et al., 2012), the 9[th] October 2014 major flood (Silvestro et al., 2015b) led to a
peak flow around 1100-1200 $m^3/s$, the peak flow of the well-known flood on 7[th] October
1970 was estimated around 1100 $m^3/s$ (Rosso, 2014).
We considered the configuration with RSs=4 km and RSt=3 hrs in order to account for
spatial and temporal uncertainty of rainfall pattern and to give a certain variability to the
disaggregated rainfall fields, and to maintain a certain spatial-temporal coherence between
RSs and RSt (Rebora et al., 2006b); we extracted the hydrographs that lead to the peak
flows with 10, 25, 50, 75, 90 percentiles (hereafter perc10 to perc90), they are reported in
Figure 6.
Figure 6
The time series furnish important information. Firstly they confirm the severity of the
possible streamflow scenarios (consider that given the current structural condition of the
riverbed the flooding threshold is around 700 $m^3/s$); secondly they evidence that the
flooding would have occurred between 12:00 and 16:00 UTC (14:00 to 18:00 local time)
when the potential risk for human lives and goods were very high. In fact during that time
window the city is in full activity: there is a lot of traffic due to people that uses means of
transport for work, the shops and stores are open, kids and children exit from school.
**4.2   Hydraulic model validation**
The extent of hazard map was estimated using the hydraulic model Telemac-2D. The
basic static input data used by Telecam-2D is a Digital Elevation Model (DEM), in this
application a DEM with 1 m spatial resolution acquired by Light Detection And Ranging




(LIDAR) technology was used; DEM information was integrated with a detailed
description of the Bisagno riverbed derived by survey measurements done between
August 2012 and June 2013. The aforementioned data were used to describe the topology
of the area of the city of Genova affected by the Bisagno creek flooding events. The
hydraulic model was set and calibrated to reproduce historical flooding especially the one
occurred the 9[th] October 2014 (Silvestro et al., 2015b), for this latter a lot of data are in
fact available together with a large number of field measurements that allowed to well
estimate the magnitude of the flood in terms of both water level and extent (Figure 7).
The final setting of the model allows a good reproduction of the field post-event
measurements, some mismatches are present and they are due to a non perfect
reproduction of the real altitudes by the DEM in some areas, and by the fact that some
features (for example basements) cannot be described with high detail but only in a
parametric way.
Figure 7
**4.3  Hazard mapping and damage estimation**
This exposure dataset and the entire damage computation methodology presented in section
3.3 were validated referring to a recent urban flash flood, which occurred in October 9th,
2014 in Genoa (Silvestro et al., 2015b). In this event hazard and exposure-vulnerability
models were computed separately and validated against observations and claims. As showed
in paragraph 4.2 the maximum water depth values obtained by the hydraulic model were
compared and validated with flood marks collected in the aftermath of the flood as described
in section 4.2 (Figure 7). The total simulated damage was then compared and validated across
the official damage assessment obtained through citizen claims and municipal authorities





surveys (Trasforini et al., 2015). In that study, over 3000 refund requests for flood damage
were processed and georeferenced, aggregated at building and neighbourhood scale to
validate computed losses.
It must be remarked that damage at building structure and content does not represent the
whole damage reported during the event. A relevant portion of total damage was due to cars
parked in private and public parking and along the streets, to transport facilities (roads and
train station), public sewage systems. These contributions are not accounted in the presented
analysis.
The five streamflow scenarios identified in paragraph 4.1 (scenarios perc 10 to perc 90)
were used as input to Telemac-2D and then the methodologies described in section 3.3
were applied to estimate the damage and the affected population.
An important hypothesis that was done and that needs to be noticed is related to the point
where the flooding starts along the riverbed. It is in fact assumed to be constant for all the
scenarios and coincident with the flooding point occurred during the benchmark event ($9^{th}$
October 2014 flash flood) used for validation, this is not rigorously correct but we needed
to do this assumption for different reasons. Some information were not available for an
area largest than the considered one, we refer in particular to the high resolution DEM and
some data to carry out the damage estimation. All this leads to an underestimation of the
total flooding area because the areas nearby the river branch upstream the considered
point are not accounted.
The results are presented in Figure 8 to 10 where hazard maps are shown together with
economic damage at building scale.
As can be easily seen the flooding affects a large heavily urbanized area, where several
stores, offices, retail trade activities, schools and residential buildings are placed. The



extent of the affected area weakly changes between perc10 to perc90 scenarios because of
the topology of the city; anyway the water level in various areas changes dramatically
increasing even of 2-3 meters. This is due to the increasing of flooding volumes and their
accumulation on the depressed areas. This occurrence clearly leads to a different impact in
terms of damage to goods and to a different level of risk for the lives of citizens.
In table 2 and 3 the estimation of economic damage is reported for each flooding scenario
compared with the damage estimated for the 9[th] October 2014 flash flood, used as
benchmark, during which a peak flow that correspond to a $100 < T < 200$ yrs was
registered. Results are reported both as absolute values and percentage values and
separating the damage to the structures from damage to the content. It is impressive that
the total damage ranges between about 141 Mln € and 232 Mln €, that in percentage
means a range between 140 and 231 % of the 2014 event. Even the Perc10 scenario leads
to a larger amount of damages in respect to the benchmark event notwithstanding the peak
flows are comparable; this is probably due by a larger overbanking volume.
Table 2
Table 3
Table 4 reports the total affected population and their distribution on the areas at different
level of risk. Population was distributed according to a day-time scenario (the hypothetical
event would have occurred between 14:00 and 18:00 local time), considering that people
can be found not only in dwellings but also in commercial and industrial buildings,
schools, etc. (see paragraph 3.3.2 "Exposure-population")
Table 4





Figure 11 to 13 show the maps with zones at different hazard level together with the
affected population assigned to each building, while table 4 reports the total affected
population and its distribution in zones with different level of hazard.
The total population that can be potentially affected by flooding is quite high (almost
19000 people) and does not significantly change from a scenario to another. This is due to
the fact that the extension of the inundated area does not change significantly. Clearly the
percentage of people that can found themselves in areas at high or very high level of risk
increases with the hazardousness of the scenarios (from Perc10 to Perc90), because of the
different water levels and different flow velocities. This fact is evidenced both by Figures
11 to 13 and by the table 4.
**5    Discussion and conclusion**
The presented work analyses the consequences of a hypothetical but realistic event in
Genova city located in correspondence of the mouth of Bisagno Creek, Liguria Region,
Italy.  This approach aims at quantifying impacts of possible real events in a "worst case"
perspective. This is accomplished considering the rainfall field occurred during a real
flash flood event at about 50 km of distance and transferring it over the target catchment
following a robust and novel methodology based on the work presented in Silvestro et al.
(2015b). The motivations that drove this kind of analysis can be found reading various
papers (Buzzi et al., 2013; Delrieu et al., 2006;Rebora et al., 2013; Silvestro et al., 2012;
Silvestro et al., 2015b) which show that this kind of very intense rainfall structures can
potentially strike, more or less indifferently, a large portion of the Liguria Region Coast.
The rainfall field was used as input to a Flood Forecast Framework made by a
downscaling model and an hydrological model in order to account for uncertainties related



to the spatial and temporal structure of the rainfall pattern and to generate an ensemble of
possible streamflow scenarios; a subset of these streamflow scenarios was then used to
feed a hydraulic model in order to simulate the hazard maps. These latter are then used to
estimate the damages with a proper methodology developed within the RASOR FP project
(Rudari 2015; Koudogbo et al., 2014) .
The results of the experiments can be summarized as follows:
1) The rainfall event lead to a very low frequent and extreme flood event on Bisagno

8        creek, the peak flow at the section Passerella Firpo (located in the city of Genova) is

9        around 1400-1600 m³/s that correspond to a return period T larger than 200 yrs.

2) Peak flows of the aforementioned magnitudes are realistic and possible even if in

11       living memory they never occurred. This is not a commonplace result. In fact,

12       generally, these high flow values (T>200 yrs) are the result of statistical analysis of

13       observed/simulated annual maxima time series with reduced length N (with N < 50-

14       100 values) so very uncertain. The experiment generates such streamflow magnitude

15       using a real rainfall event and considering a realistic soil moisture as initial condition

16       of the study area.

17   3) The flooding of Bisagno creek in correspondence of the city of Genoa leads to a large

18       inundation area with water level even higher of 2-3 meters in the centre of the city.

19       The large volume of flooding produces large accumulation in the streets especially in

20       depressed area

21   4) The over banking occurs between 12:00 and 16:00 utc (14:00 to 18:00 local time)

22       which is a time window really dangerous with a large number of persons that can be

23       potentially affected by the inundation



5) The estimated damages to the structures and their content is between 141 and 232 Mln of euro that means 140 to 231 % of the benchmark event, that was caused by a peak flow with 100 yrs < T< 200 yrs.

6) The population potentially affected is roughly between 17000 and 19000 units, with a distribution in the areas at high and very high hazard level which ranges between 3600 and 7700 units. This is a conservative estimation since the applied methodology does not completely account for people that live out of the affected area, but can access the area during their daytime activities.

These results show how devastating could be an event of such a magnitude and they highlight the need of augmenting the resilience of the city and of its population. Sophisticated and state of the art Early Warning Systems (EWS) as well as nowcasting techniques (Silvestro et al., 2011; Berenguer et al., 2005) are already operational in the study area as well a Civil Protection system that is able to act on the territory (Brandolini et al., 2012). Anyway we have to consider that EWS can fail especially in the exact localization of the event (Silvestro et al., 2015b, Buzzi et al., 2013) and that a Civil Protection system is effective when the population is able to translate the Alert and Warning messages in tangible behaviors and actions. The preparedness and correct information of the population is a basic prerequisite to save lives and try to reduce the loss of goods: people (especially who live or work in areas at high risk) should know exactly how to behave in case of event avoiding such actions that increase their risk. Moreover, even if in the case of a (purely hypothetical) perfect EWS, which enables Civil Protection to issue prompt alert messages and saves all the population, the level of damage would be




huge anyway, causing large problems to the economy of city. With this respect,
retrofitting measure aimed to reduce vulnerability (i.e. some small investments such as
rails for stoplogs) can be useful in order to reduce the damages, especially in those areas
were water level do not reach very high values. These interventions can be really effective
until structural measures are completed and they can be useful to manage the residual risk
once structural interventions are done. In the specific case, a series of structural measures,
designed to avoid flooding driven by peak flows with T≤200 yrs are planned for the next
years.
(http://cartogis.provincia.genova.it/cartogis/pdb/bisagno/bisagno/documenti/PianoInterven
ti.pdf).
**Acknowledgements**
This work is supported by UE through RASOR Project (Program FP7), by the Italian National
Civil Protection Department and by the Italian Region of Liguria.





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

| LABEL | OCCUPANCY CLASS | |
|---|---|---|
| RES1 | Single Family Dwelling | RESIDENTIAL |
| RES2 | Mobile Home | |
| RES3A | Multi Family Dwelling - Duplex | |
| RES3B | Multi Family Dwelling – 3-4 Units | |
| RES3C | Multi Family Dwelling – 5-9 Units | |
| RES3D | Multi Family Dwelling – 10-19 Units | |
| RES3E | Multi Family Dwelling – 20-49 Units | |
| RES3F | Multi Family Dwelling – 50+ Units | |
| RES4 | Temporary Lodging | |
| RES5 | Institutional Dormitory | |
| RES6 | Nursing Home | |
| COM1 | Retail Trade | COMMERCIAL |
| COM2 | Wholesale Trade | |
| COM3 | Personal and Repair Services | |
| COM4 | Business/Professional/Technical Services | |
| COM5 | Depository Institutions (e.g. bank) | |
| COM6 | Hospital | |
| COM7 | Medical Office/Clinic | |
| COM8 | Entertainment & Recreation (e.g. restaurants and bar) | |
| COM9 | Theatres | |
| COM10 | Parking | |
| IND1 | Heavy | INDUSTRIAL |
| IND2 | Light | |
| IND3 | Food/Drugs/Chemicals | |
| IND4 | Metals/Minerals Processing | |
| IND5 | High Technology | |
| IND6 | Construction | |
| AGR1 | Agriculture | AGRICULTURE |
| REL1 | Church/Membership Organization | RELIGION/NON-PROFIT |
| GOV1 | General Services | GOVERNMENT |
| GOV2 | Emergency response | |
| EDU1 | Schools/Libraries | EDUCATION |
| EDU2 | Colleges/Universities | |
| COM1+RES | Residential with retail at ground floor | MIXED |
| COM5+RES | Residential with bank at ground floor | |
| COM8+RES | Restaurant and bar | |

4  Table 1 Original HAZUS building occupancy classes (grey) and derived mixed occupancy
5  classes (yellow).





| | Perc10 | Perc25 | Perc50 | Perc75 | Perc90 | 2014 event |
|---|---|---|---|---|---|---|
| Economic Damage at structure [Mln €] | 42.7 | 53.7 | 59.3 | 67.3 | 73.6 | 29.7 |
| Economic Damage at Content [Mln €] | 97,9 | 121.9 | 134.5 | 148.6 | 158 | 70.4 |
| Total Damage [Mln €] | 140.6 | 175.6 | 193.8 | 211.9 | 231.6 | 100.1 |

3  Table 2: economic damage estimated for the considered flooding scenarios compared with

4  damage estimated for the event on 9[th] October 2014.



| | Perc10 | Perc25 | Perc50 | Perc75 | Perc90 | 2014 event |
|---|---|---|---|---|---|---|
| Economic Damage at structure in respect 2014 event [%] | 144% | 181% | 200% | 227% | 248% | 100% |
| Economic Damage at content with respect to 2014 event [%] | 139% | 173% | 191% | 212% | 224% | 100% |
| % Total Economic Damage with respect to 2014 event [%] | 140% | 175% | 194% | 212% | 231% | 100% |

1    Table 3: Ratio between damage estimated for the considered flooding scenarios and

2    damage estimated for the event on 9[th] October 2014





| Scenario | Total [units] | Low Hazard [units] | Moderate Hazard [units] | High Hazard [units] | Very High Hazard [units] |
|---|---|---|---|---|---|
| Perc10 | 17360 | 3085 | 10705 | 3520 | 50 |
| Perc25 | 18255 | 2390 | 11175 | 4400 | 290 |
| Perc50 | 18440 | 2140 | 10475 | 5195 | 630 |
| Perc75 | 18645 | 1975 | 10005 | 5675 | 990 |
| Perc90 | 18805 | 1890 | 9205 | 6360 | 1350 |

1 Table 4: population potentially affected by the different flooding scenarios and their

2 distribution on the zones with different levels of risk. The total is estimated summing the

3 population of the Low, Moderate, High and very High Risk zones.


**7    Figures**

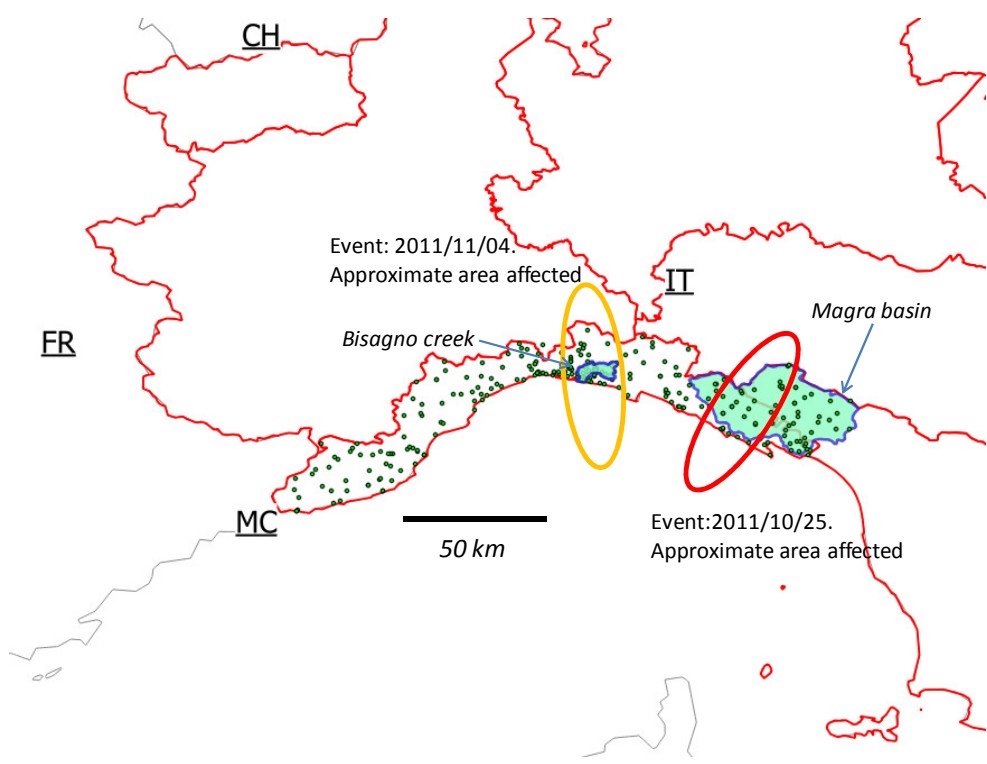

Figure 1: Main areas stroke by the two intense events occurred between October and
November 2011 (red and yellow ellipses). The watermarks of the Bisagno creek and of the
Magra basin are reported in blue, the green dots are the rain gauges of the regional network.





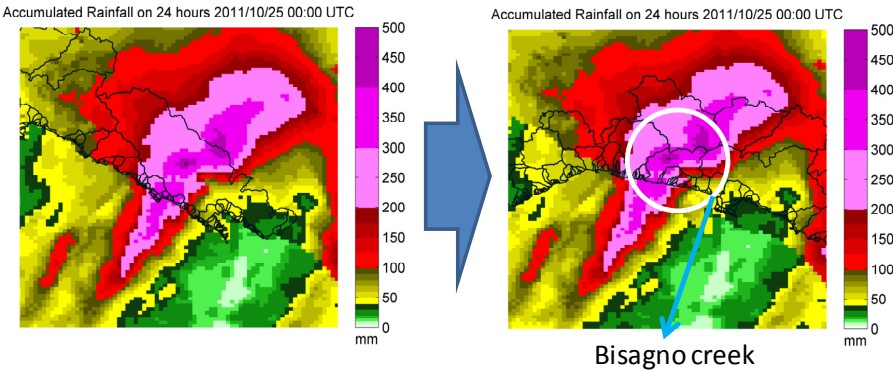

2   Figure 2: 2011/10/25, accumulated rainfall on 24 hours. Left panel, observed rainfall field;

3   right panel, hypothetical rainfall field obtained by the rigid translation of the observed rainfall

4   field from the original position to the Bisagno creek.



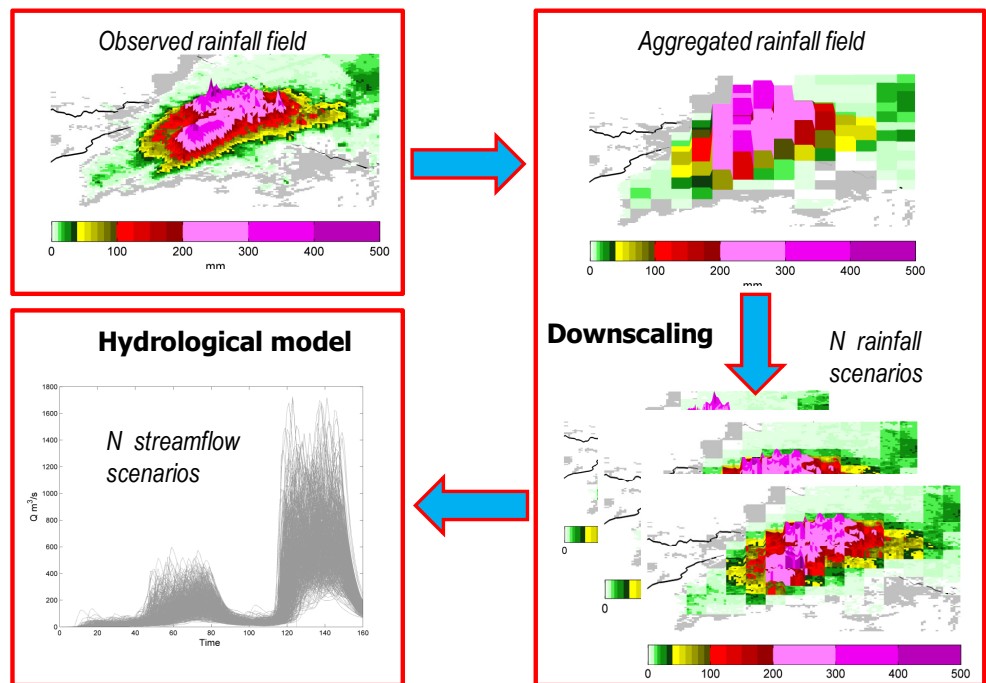

2    Figure 3: Schematization of the Flood Forecast Framework made by a downscaling model

3    and a hydrological model. In this application the rainfall field is the one reported in figure

4    2.





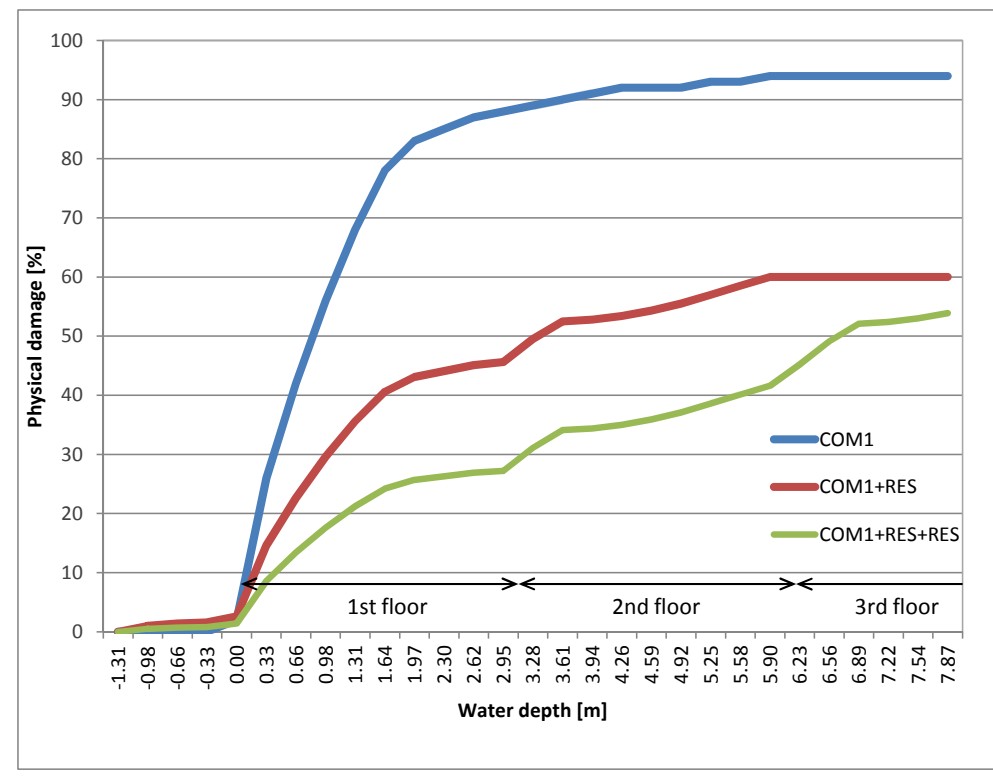

2    Figure 4: Comparison between water depth – damage curves for content: retail trade (COM1)

3    building[blue], mixed retail trade (COM1) at first floor & RES at second floor[red], mixed

4    retail trade (COM1) at first floor & residential (RES) at second and third floor[green].



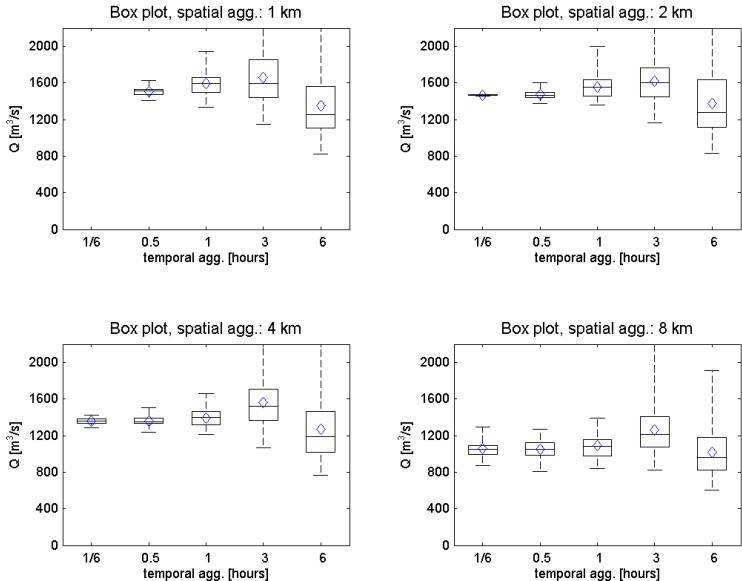

Figure 5: Passerella Firpo reference section, Area: 93 km$^2$. Box plot of the peak flow
generated by the FFF. On Y axis the peak flow is reported, on X axis the temporal
aggregation scales (RSt) are reported. Diamonds represent the peak flow of the reference
hydrograph. Each sub-panel shows results for a different spatial aggregation scale (RSs).





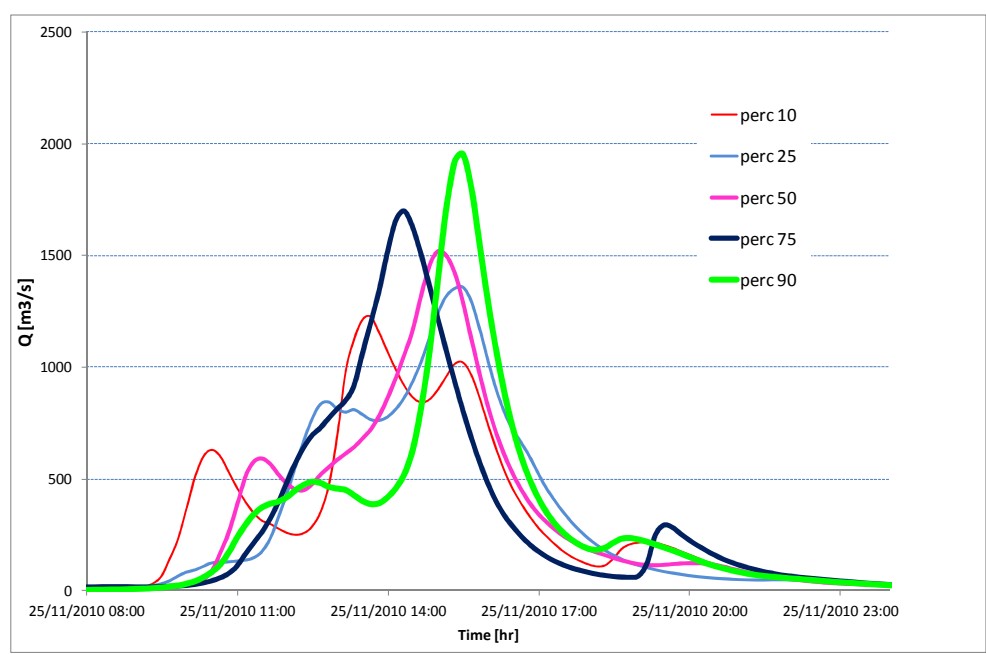

2      Figure 6. Streamflow scenarios derived by RSs=4 km and RSt=3 hrs. The hydrographs that

3      lead to the peak flows with 10, 25, 50, 75, 90 percentiles were extracted.



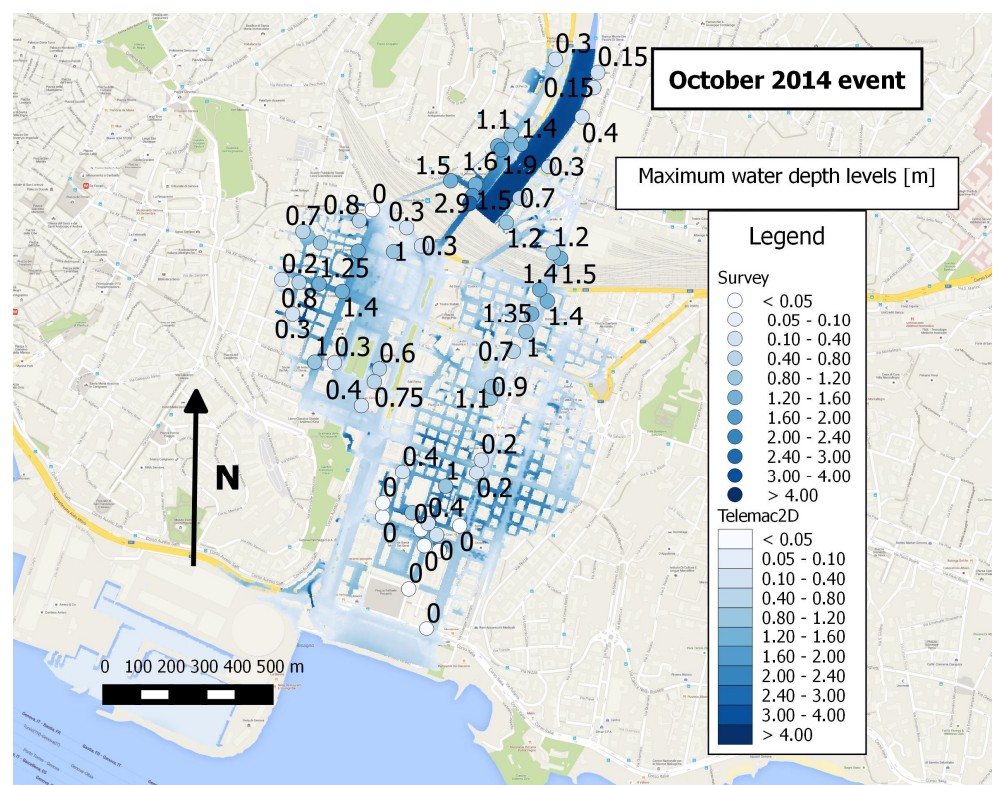

Figure 7. Center of Genova city. Flood occurred on 9$^{th}$ October 2014. Comparison of the

maximum flooding extent obtained through Telemac-2D and the field observations. The

model was set in order to obtain the best fit between modeling and observations.



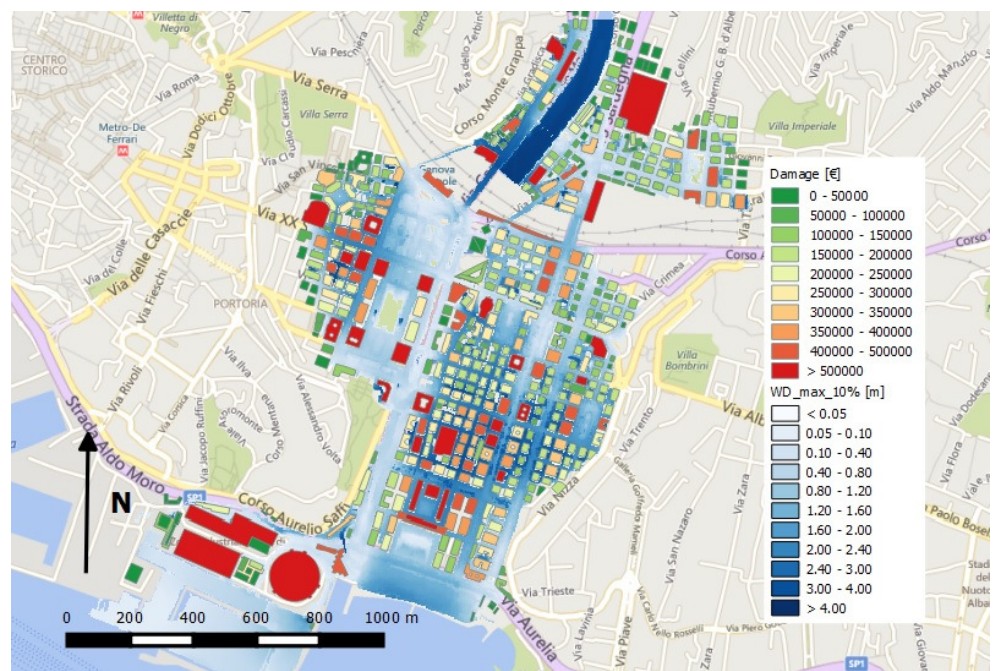

3 Figure 8: Perc10 scenario, inundation map and damage estimation. In blue scale the water

4 level is reported. The damage is estimated at building scale in euro, the color scale ranges

5 from low damage (green) to high damage (red).




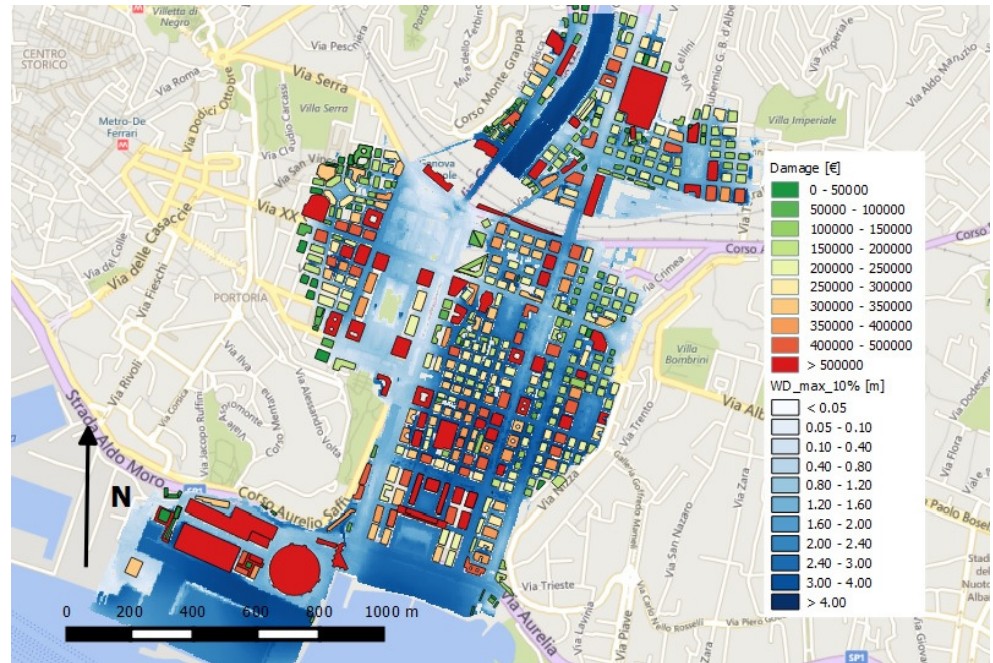

Figure 9: Perc50 scenario, inundation map and damage estimation. In blue scale the water level is reported. The damage is estimated at building scale in euro, the color scale ranges from low damage (green) to high damage (red).





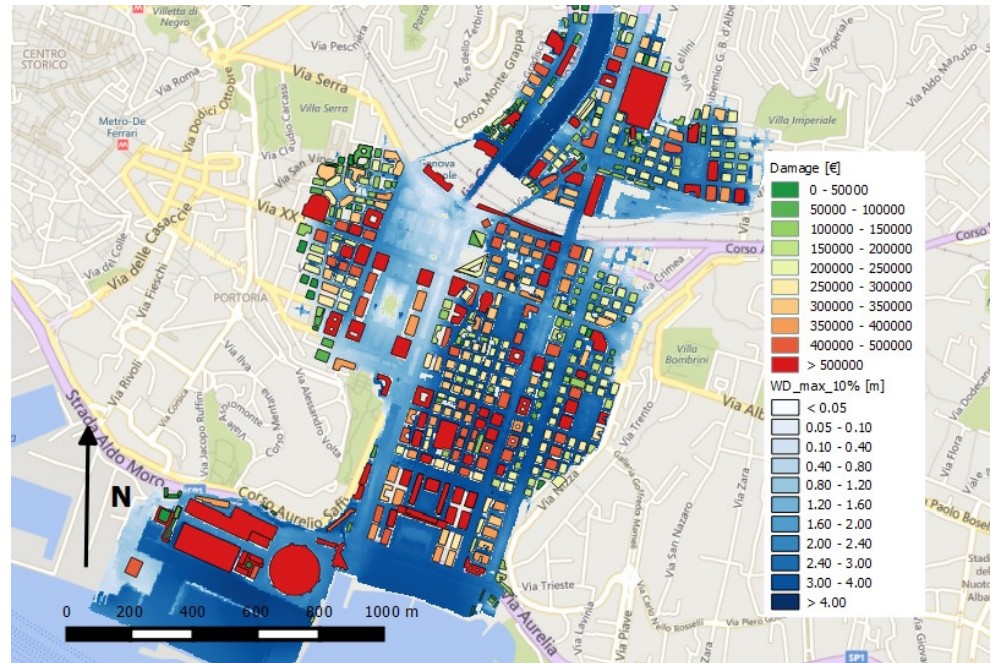

Figure 10: Perc90 scenario, inundation map and damage estimation. In blue scale the water level is reported. The damage is estimated at building scale in euro, the color scale ranges from low damage (green) to high damage (red).





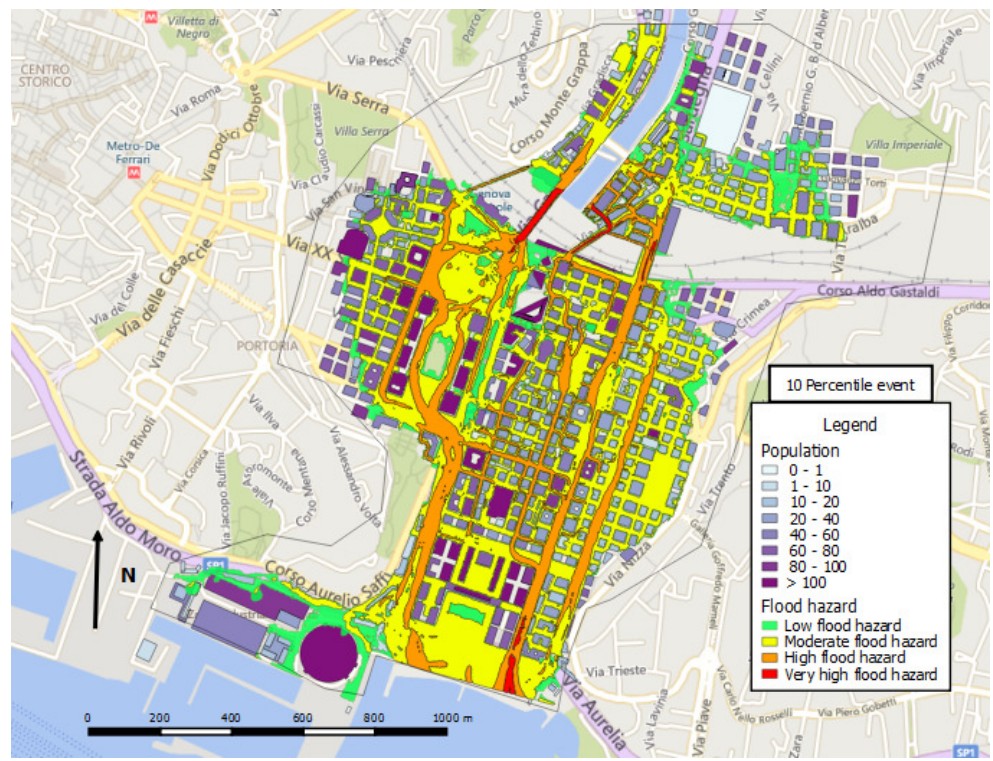

2    Figure 11: Perc10 scenario, hazard level map compared with population potentially involved

3    assigned to each building.



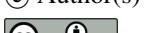

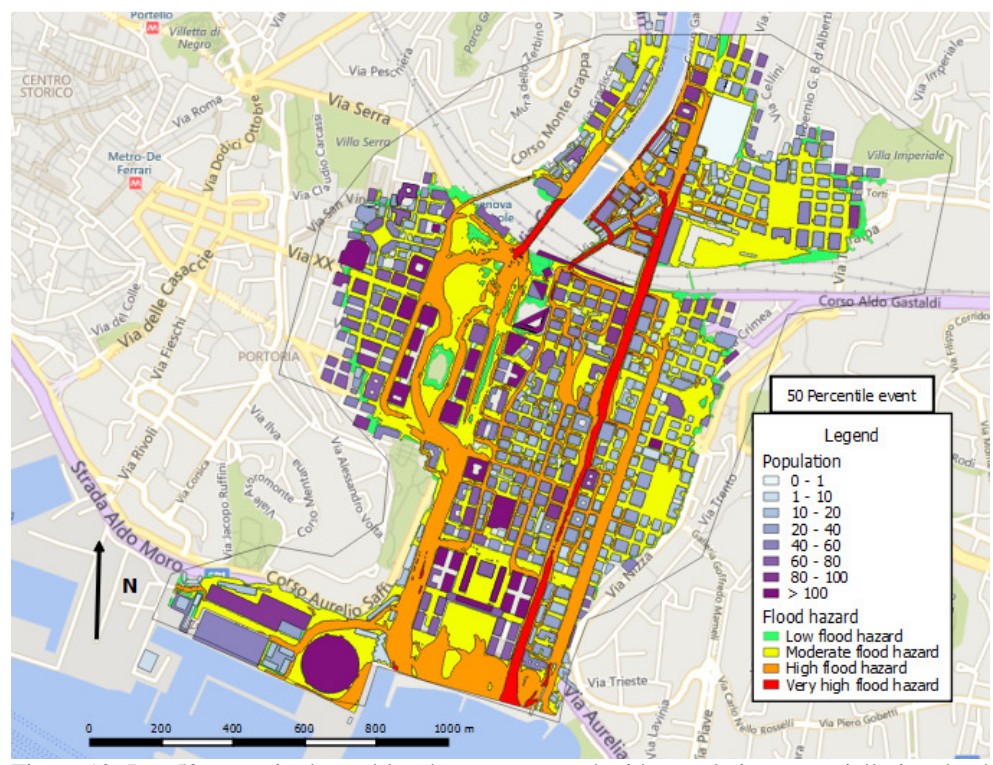

2   Figure 12: Perc50 scenario, hazard level map compared with population potentially involved

3   assigned to each building.





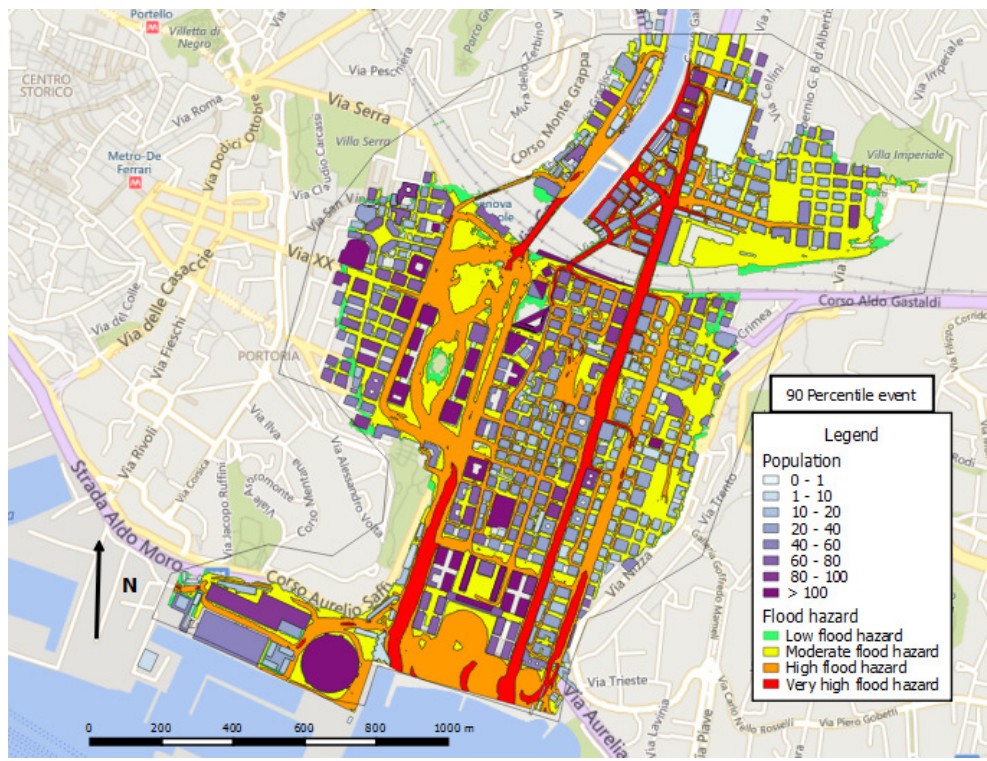

2     Figure 13: Perc90 scenario, hazard level map compared with population potentially involved

3     assigned to each building.

