# Peer review of "What if the 25th October 2011 event that stroke Cinque Terre (Liguria) had happened in Genova, Italy? Flooding scenarios, hazard mapping and damages estimation."

_Natural Hazards and Earth System Sciences, 2016_

## Referee Comment (RC1) · G. Pegram (Referee) · 6 May 2016

This is a carefully and usefully constructed paper that combines high-end methodologies and pragmatic strategies to evaluate the risk and cost of flooding of an important Italian city. The clever part is the translation of a nearby storm of greater magnitude and risk level than that experienced at the same time in Genoa, to see how people and structures in a large city would be affected and at what cost. The technical passages report a study that is state-of-the-art in terms of the care and detail that has been

used in the development of the combined methodology. The reasoning and layout of the paper is exemplary. I have 2 requests: change 'stroke' to 'struck' in the title and elsewhere; correct some of the grammar in a very few places.

After the paper has been polished, I recommend that it is published in NHESS.

Geoff Pegram 06 May 2016

---

## Referee Comment (RC2) · M.C. Llasat (Referee) · 22 May 2016

General Comments

The article deals with an interesting exercise to analyse the potential impacts of a big flood event in Genoa. This flood event has been created putting the rainfall field of an event recorded on October in a near region, on the main catchment of Genoa stroke by another catastrophic flood event recorded some weeks after, on November. It is a very complete work: starting from the hydrometeorological chain to model the event, it ends with a methodology to estimate the damages. Consequently, although it refers to

a specific region, the proposed methodology will be useful to be reproduced in other regions.

Abstract: The abstract should be rewritten. In its present form it seems more an Introduction than an abstract. Parts of the abstract could be moved to the Introduction. After a very short introduction (one or two sentences) the abstract should be focused on the objective of the paper, cases study and methodology and results.

Section 1: Some parts of the Introduction should be moved to the methodology: P.4 Lines 15-23, P.5 Lines 1-8.

It will be useful to have a better understanding of both events and the reason that justifies the exercise, to include some explanation about the October event that affected Cinqueterre in comparison with the November event that affected Genoa (maximum cumulated rainfall and hourly intensity, damages, relative discharge,. . .)

Section 2: Please, tell the main features of the Magra basin

Section 3: Please, indicate where the Passerella Firpo level gauge is (Bisagno creek? Magra basin?). Could you illustrate the example explained in page 13 with a figure?

Conclusions and Discussion They constitute a good synthesis and promote a reflection for the authorities of Genoa

Please also note the supplement to this comment:
http://www.nat-hazards-earth-syst-sci-discuss.net/nhess-2016-125/nhess-2016-125-RC2-supplement.pdf
* * *
[Figure]

**Supplement:**

**Review of the paper "What if the 25th October 2011 event that stroke Cinque Terre (Liguria) had happened in Genova, Italy? Flooding scenarios, hazard mapping and damages estimation. submitted to NHESS by**

**Francesco Silvestro, Nicola Rebora, Lauro Rossi, Daniele Dolia, Simone Gabellani, Flavio Pignone, Eva Trasforini, Roberto Rudari, Silvia De Angeli1, Cristiano Masciulli**

**General Comments**

The article deals with an interesting exercise to analyse the potential impacts of a big flood event in Genoa. This flood event has been created putting the rainfall field of an event recorded on October in a near region, on the main catchment of Genoa stroke by another catastrophic flood event recorded some weeks after, on November. It is a very complete work: starting from the hydrometeorological chain to model the event, it ends with a methodology to estimate the damages. Consequently, although it refers to a specific region, the proposed methodology will be useful to be reproduced in other regions.

**Abstract:**
The abstract should be rewritten. In its present form it seems more an Introduction than an abstract. Parts of the abstract could be moved to the Introduction. After a very short introduction (one or two sentences) the abstract should be focused on the objective of the paper, cases study and methodology and results.

**Section 1:**
Some parts of the Introduction should be moved to the methodology: P.4 Lines 15-23, P.5 Lines 1-8.

It will be useful to have a better understanding of both events and the reason that justifies the exercise, to include some explanation about the October event that affected Cinqueterre in comparison with the November event that affected Genoa (maximum cumulated rainfall and hourly intensity, damages, relative discharge,…)

**Section 2:**
Please, tell the main features of the Magra basin

**Section 3:**
Please, indicate where the Passerella Firpo level gauge is (Bisagno creek? Magra basin?). Could you illustrate the example explained in page 13 with a figure?

**Conclusions and Discussion**
They constitute a good synthesis and promote a reflection for the authorities of Genoa

**Specific comments**

P.2, Line 10: Separate the words "approach combines"
P.3. Line 2: There are some parenthesis and "O" that should be deleted.
P.3. Line 3: To define flash floods, I think it would be better to use other references, like www.nws.noaa.gov; Gaume and Borga, 2008 or Borga et al., 2008.
- Gaume, E., Borga, M., 2008. Post-flood field investigations in upland catchments after major flash floods: proposal of a methodology and illustrations. J. Flood Risk Manag. 1, 175–189.

- Borga, M., Gaume, E., Creutin, J.D., Marchi, L., 2008. Surveying flash flood response: gauging the ungauged extremes. Hydrol. Process. 22 (18), 3883–3885.

P.3. Line 20: Put a point between "authorities" and "Rebora"

P.3. Line 22: Put a point between "question" and "in fact"

P.4. Line 9: Put a point between "data" and "Buzzi"

P.4. Line 11: Put a point between "genesis" and "in addition" and a comma after this.

P.4. Line 19: Add "it" is downscaled

P.4. Line 21: Put a point between "experiment" and "in fact"

P.4. Line 22: "allows accounting"

P.5. Line 2: "…large scale"

P.5. Line 21: Genova is also in Liguria. Then, it is better to write only (Liguria, Italy) after "Genova".

P.7. Line 20: Substitute "here" by "where"

P.8. Lines 13-18: DV? RS? AF?

P.8. Line 3. Please indicate for which hourly interval this precipitation was recorded. Which was the duration of the total event? When the precipitation event is moved to the Genoa catchment, is the time distribution the same?

P.8. Line 11 and P.9. Lines 6-10. If the RainFARM product is a downscaling product, why is necessary to aggregate the radar data and disaggregate it posteriorly?

P.9. Line 1: RVs?

P.9. Line 9: "Continuum"

P.10. Line 7: "hazards. It allows easily updating"

P.10. Line 18: EO?

P.12. Line 7: "2005; Freire, 2010)"

P.12. Line 10: Delete the initial letters of the authors' name.

P.13. Line 3: Please, include a reference for the HAZUS-MH database.

P.14. Line 16: Please, use the super index format for the square meters.

P.15. Line 5: Please, delete the initials ML.

P.15. Lines 8-10: It seems that the verb lacks

P.15. Line 15: Delete the word "where" in the parenthesis.

P.15. Line 20: Replace $0<h<0.2$ by $h<0.2$; replace $h \geq 0.2$ m and $h <0.5m$ by $0.5>h\geq 0.2m$. Is there any reference for these thresholds? Please, define h (I suppose it is the water level in the inundated street)

P.16. Line 1: Replace $h > 0$ m and $h<0.2m$ by $h<0.2m$; Replace $h >0.2m$ 1 and $h <0.5m$ by $0.2<h<0.5$ m.

P.16. Line 20: Replace yrs. by y.

P.16. Line 22-P.17. Lines 1-3: Please, indicate in which gauge stations and regions the different peak flows were recorded. The 7[th] October event is not necessary "well-known for the reader. Please add a parenthesis with some information about it that justifies its importance.

P.17. Line 8: " as they are reported…"

P.17. Line 21: Telemac-2D is a part of the Telemac-Mascaret?

P.17. Line 22: Replace Telecam by Telemac

P.17. Line 22: Replace the comma before "in" by a dot.

P.18. Line 6: Replace the comma before "for" by a dot.

P.18. Line 10: Replace the comma before "some" by a dot

P.19. Lines 9-24: The paragraph is indented.

P.19. Lines 16-18: The meaning of the sentence "some information…estimation" is not clear. Please, rewrite it.

P.20. Line 11: Substitute Mln € by M€ I suppose that these quantities refer to the simulated event, but, please, remind it to the reader. Do the same change in p. 23.

P.21. Line 6: I suppose that the extension of the inundated area does not change due to the orography, but it will be better to add a comment to justify it.

P.22. Line 7: It will be better to say "the hypothetical rainfall event…" or something similar.

P.22. Line 21: UTC in capital letters

P.22. Line 22: Replace persons by people.

P.24. Lines 9-10. Remove this reference to the cartogis, or cite it correctly.

**References**
P.25. Line 13. It is Diezma not Diesma.
I would suggest you to include the papers from Fiori et al 2014 (Atmospheric Research) and Hally et al 2015 (NHESS) in the references

Figure 1: Please, show where the city of Genova is and the position of the radar.

---

## Author Comment (AC1) · 30 Jun 2016

**What if the 25[th] October 2011 event that struck Cinque Terre (Liguria) had happened in Genova, Italy? Flooding scenarios, hazard mapping and damages estimation.**

**Francesco Silvestro[1*], Nicola Rebora[1], Lauro Rossi[1], Daniele Dolia[1], Simone Gabellani[1], Flavio Pignone[1], Eva Trasforini[1], Roberto Rudari[1], Silvia De Angeli[1,2], Cristiano Masciulli[3]**

*Dear Editor and Reviewers,*

*In the following we report the editor and reviewers comments with our replies in italic.*

*Since most of the modifications are text modifications we report, after the responses, a new version of the manuscript. We based the improvements on the comment's answers.*

*Along the text we highlighted in yellow the parts of the manuscript where major changes have been applied. Along the text even other modifications have been applied (for example: text modifications, figures modification, …etc).*

*We hope that the manuscript is now publishable in NHESS but we are open to introduce further improvements.*

*Best regards.*

**Referee 1**

I have 2 requests: change 'stroke' to 'struck' in the title and elsewhere; correct some of the grammar in a very few places.

*We changed stroke with struck as requested and we revise the grammar and typing. If it is necessary we will revise the manuscript with the help of a native English speaker.*

**Referee 2**

Abstract: The abstract should be rewritten. In its present form it seems more an Introduction than an abstract. Parts of the abstract could be moved to the Introduction. After a very short introduction (one or two sentences) the abstract should be focused on the objective of the paper, cases study and methodology and results.

*The abstract was rewritten as suggested by the reviewer and part of its first version was moved on the text of section 1. (pgg 3-4 lines 22-3)*

Some parts of the Introduction should be moved to the methodology: P.4 Lines 15-23, P.5 Lines 1-8.

*We moved part of the introduction to section 3. We left on section 1 only a brief explanation of the method we applied (pgg 9 lines 3-12)*

It will be useful to have a better understanding of both events and the reason that justifies the exercise, to include some explanation about the October event that affected Cinqueterre in comparison with the November event that affected Genoa (maximum cumulated rainfall and hourly intensity, damages, relative discharge,…)

*We added more details of the two events and a new figure in section 1 as requested (pgg 4 lines.4-15)*

[Figure]

Section 2: Please, tell the main features of the Magra basin

*In the presented application we do not do any hydrological evaluation on Magra*
*basin and the focus is not the comparison of the effects of the two events. It appears*
*to us a little bit out of the scope of the paper to insert in section 2 the description of*
*that basin since there we described Bisagno creek which is the target area. In section*
*1 we mention Magra basin and its drainage area(pgg 4 line 16).*

*Anyway if the reviewer retains necessary inserting more characteristics we are open*
*to do this.*

Please, indicate where the Passerella Firpo level gauge is (Bisagno creek?
Magra basin?).

*We indicated that it refers to Bisagno creek (pgg 9 line 4)*

Could you illustrate the example explained in page 13 with a figure?

*The requested figure was added*

[Figure]

*Figure 5: An example of mixed-use curve definition. The green curve corresponds to*
*the flood vulnerability function for the content of a 2-storey building, with mixed*
*commercial and residential use: specifically retail trade at ground floor and residential*

*at first floor. It is obtained by combining the 1-storey curve for generic residential (in red) with the 1-storey curve for retail trade (in blue). The yellow section of the graph represents the average height of the ground floor. The left part of the mixed curve is obtained by re-scaling the blue curve. For higher values of water level, the function increases proportionally to the values that the residential curves assumes (red) in the left part of the graph.*
*It is assumed that the value of the ground floor is equal to 60% of the whole (2-storeys) building.*

**Specific comments**

P.2, Line 10: Separate the words "approach combines"

*The sentence was moved and corrected*

P.3. Line 2: There are some parenthesis and "O" that should be deleted.

*We changed the sentence with "….and fast response of catchments with O(Area) 100 to 103 km2 (Quevauviller, 2014)."*

P.3. Line 3: To define flash floods, I think it would be better to use other references, like www.nws.noaa.gov; Gaume and Borga, 2008 or Borga et al., 2008.

*We added one of the references suggested by the reviewers*

P.3. Line 20: Put a point between "authorities" and "Rebora"

P.3. Line 22: Put a point between "question" and "in fact"

P.4. Line 9: Put a point between "data" and "Buzzi"

P.4. Line 11: Put a point between "genesis" and "in addition" and a comma after this.

*Ok, done*

P.4. Line 19: Add "it" is downscaled

P.4. Line 21: Put a point between "experiment" and "in fact"

P.4. Line 22: "allows accounting"

*Ok, done*

P.4. Line 22: "allows accounting"

P.5. Line 2: "…large scale"

P.5. Line 21: Genova is also in Liguria. Then, it is better to write only (Liguria, Italy) after "Genova".

P.7. Line 20: Substitute "here" by "where"

*Ok, done*

P.8. Lines 13-18: DV? RS? AF?

*We changed the text trying to make it clearer.*

P.8. Line 3. Please indicate for which hourly interval this precipitation was recorded. Which was the duration of the total event? When the precipitation event is moved to the Genoa catchment, is the time distribution the same?

*We changed the text: "The rainfall field occurred from the 00:00 UTC of 25th October 2011 to the 00:00 UTC of 26th October 2011"*

*The time distribution is one of the variables of the downscaling procedure.*

*When the temporal aggregation scale is 60 minutes, the time distribution is the same of the observations.*

*When the temporal aggregation scale is 360 minutes, the time distribution is lost inside the 6 hours length aggregation time windows and it is generated by the downscaling algorithm.*

P.8. Line 11 and P.9. Lines 6-10. If the RainFARM product is a downscaling product, why is necessary to aggregate the radar data and disaggregate it posteriorly?

*Because in this way we can analyze the effects of changing the temporal and spatial structure of the precipitation but maintaining always the total volume.*

*This effect should be clear looking at Figure 5 (7 in the new version)*

*Example:*

*Spatial agg. 2km and Temporal agg: 1/6 hour. This is (almost) a rigid translation of the event in fact the streamflow scenarios are really similar*

*Spatial agg. 2km and Temporal agg: 3 hour. The temporal structure changes (not the total volume) and the streamflow scenarios are different*

P.9. Line 1: RVs?

*Mistake. Corrected with AFs*

P.9. Line 9: "Continuum"

*Corrected*

P.10. Line 7: "hazards. It allows easily updating"

*Ok*

P.10. Line 18: EO?

*Corrected*

P.12. Line 7: "2005; Freire, 2010)"

*Corrected*

P.12. Line 10: Delete the initial letters of the authors' name.

*Corrected*

P.13. Line 3: Please, include a reference for the HAZUS-MH database.

*Ok inserted*

P.14. Line 16: Please, use the super index format for the square meters.

*Corrected*

P.15. Line 5: Please, delete the initials ML.

*Corrected*

P.15. Lines 8-10: It seems that the verb lacks

*The new sentence is "In order to compare possible impacts on population for different scenarios, four hazard zones (very high, high, moderate, low flood hazard) were defined based on the human instability in floodwaters."*

P.15. Line 15: Delete the word "where" in the parenthesis.

*Corrected*

P.15. Line 20: Replace $0<h<0.2$ by $h<0.2$; replace $h \geq 0.2$ m and $h <0.5$m by $0.5>h\geq 0.2$m. Is there any reference for these thresholds? Please, define h (I suppose it is the water level in the inundated street)

P.16. Line 1: Replace h > 0 m and h<0.2m by h<0.2m; Replace h >0.2m 1 and h <0.5m by 0.2<h<0.5 m.

*We changed the text as suggested and inserted the definitions of h and v some lines above :"…*

*that in flow conditions 0.5 < v < 3 m/s and 0.3 < h < 1.5 m (where v and h are the velocity and the water level in the inundated street)…".*

*The thresholds to differentiate high from very high hazard and moderate from low hazard are introduced based on experience. We modified the text in order to point out this fact "…Further thresholds (upper and lower) were introduced based on "expert judgement" in order to identify…"*

DAI P.16. Line 20: Replace yrs. by y.

*We used yrs to indicate "years" along all the manuscript since we used it in other published papers so we propose to maintain this nomenclature. If the reviewers retain this should be change we can modify it.*

P.16. Line 22-P.17. Lines 1-3: Please, indicate in which gauge stations and regions the different peak flows were recorded. The 7th October event is not necessary "well-known for the reader. Please add a parenthesis with some information about it that justifies its importance.

*In section 3.1 we stated that passerella Firpo on Bisagno creek is the reference section: "…The considered reference model section correspond to the location of Passerella Firpo level gauge on Bisagno creek, there the drainage area is 93 km2…"*

*Now we added on section 4.1: "…. Figure 5 shows the box plot of the 500 peak flows generated with FFF compared with the mean peak flow of the sample of 500 realizations represented by the blue diamonds for the reference model section on Bisagno creek..."*

*Regarding the event on 7th October 1970.*

*We removed "well-known" from the text as suggested. The importance is due to the large amounts of dead people, but probably is not important for the scope of this part of the paper. The reference in the text (Rosso 2014) describes the event.*

P.17. Line 8: " as they are reported…"

*Corrected*

P.17. Line 21: Telemac-2D is a part of the Telemac-Mascaret?

*Yes as already mentioned in section 3.2*

P.17. Line 22: Replace Telecam by Telemac

P.17. Line 22: Replace the comma before "in" by a dot.

P.18. Line 6: Replace the comma before "for" by a dot.

P.18. Line 10: Replace the comma before "some" by a dot

*Corrected*

P.19. Lines 9-24: The paragraph is indented.

*Corrected*

P.19. Lines 16-18: The meaning of the sentence "some information…estimation" is not clear. Please, rewrite it

*We changed the sentence with: "….Particularly binding was the fact that some information was available only for the considered are, we refer to the high resolution DEM and to some data needed to carry out the damage estimation."*

P.20. Line 11: Substitute Mln € by M€. I suppose that these quantities refer to the simulated event, but, please, remind it to the reader. Do the same change in p. 23.

*Corrected and mentioned that the loss refers to simulated events.*

P.21. Line 6: I suppose that the extension of the inundated area does not change due to the orography, but it will be better to add a comment to justify it.

*The new secentence is:"This is due to the fact that the extension of the inundated area does not change significantly because of the topology"*

P.22. Line 7: It will be better to say "the hypothetical rainfall event…" or something similar.

*Modified as suggested*

P.22. Line 21: UTC in capital letters

*Corrected*

P.22. Line 22: Replace persons by people.

*Corrected*

Figure 1: Please, show where the city of Genova is and the position of the radar.

*The figure 1 was modified as requested*

**What if the 25[th] October 2011 event that struck Cinque Terre**

**(Liguria) had happened in Genova, Italy? Flooding scenarios,**

**hazard mapping and damages estimation.**

**Francesco Silvestro[1*], Nicola Rebora[1], Lauro Rossi[1], Daniele Dolia[1], Simone Gabellani[1],**

**Flavio Pignone[1], Eva Trasforini[1], Roberto Rudari[1], Silvia De Angeli[1,2], Cristiano**

**Masciulli[3]**

[1]{ CIMA research foundation, Savona, Italy}

[2] {WRR Programme, UME School, IUSS-Pavia, Italy}

[3] {IREN, Genova, Italy}

Corresponding author: Francesco Silvestro mail: francesco.silvestro@cimafoundation.org

CIMA Research Foundation (www.cimafoundation.org)

University Campus, Armando Magliotto, 2. 17100, Savona, Italy

Tel. +39 019230271,      fax. +39 01923027240

**Abstract**

During the autumn of 2011 two catastrophic very intense rainfall events affected two different parts of the Liguria Region of Italy causing various flash floods. The first occurred in October and the second at the beginning of November. Both the events were characterized by very high rainfall intensities (> 100 mm/h) that persisted on a small portion of territory causing local huge rainfall accumulations (> 400 mm/6h).

Two main considerations were done in order to set up this work. The first consideration is that various studies demonstrated that the two events had a similar genesis and similar triggering elements. The second very evident and coarse concern is that two main elements are needed to have a flash flood: a very intense and localized rainfall event and a catchment (or a group of catchments) to be affected. Starting from these assumptions we did the exercise of mixing the two flash floods ingredients by putting the rainfall field of the first event on the main catchment struck by the second event that has its mouth in correspondence of the biggest city of the Liguria Region: Genova.

A complete framework was set up to quantitatively carry out a "what if" experiment with the aim of evaluating the possible damages associated to this event. A probabilistic rainfall downscaling model was used to generate possible rainfall scenarios maintaining the main characteristics of the observed rainfall fields while a hydrological model transformed these rainfall scenarios in streamflow scenarios. A subset of streamflow scenarios is then used as input to a 2D hydraulic model to estimate the hazard maps and finally a proper methodology is applied for damages estimation. This leads to the estimation of the potential economic losses and of the risk level for the people that stays in the affected area.

The results are interesting, surprising and in such a way worrying: a rare but not impossible event (it occurred about 50km away from Genoa) would have caused huge damages estimated between 120 and 230 million of euros for the affected part of the city of Genova, Italy and more than 17000 potentially affected people.

Key words: flash floods, hazard, extreme rainfall, damage estimation, risk, urban hydrology.

**1    Introduction**

Flash floods are one of the most disastrous natural hazards that affect citizens in many part of the world causing high risk for them and for their goods and activities. Many types of flash floods exist but in a great number of cases they are caused by very intense (i.e. 50-150 mm/h)

and localized rainfall events that persist on the same area for hours (i.e. 4-12 hrs) causing large accumulation of precipitation and fast response of catchments with O(Area) $10^0$ to $10^3$

$km^2$ (Gaume and Borga 2008; Quevauviller, 2014). Many authors focused on the analysis of these events, their genesis and their ground effects (Amengual et al, 2007; Barthlott and

Kirshbaum, 2013; Gaume et al., 2009; Marchi et al., 2009; Delrieu et al., 2006; Massacanad et al., 1998; Roth et al., 1996), and lot of research was carried out to improve their predictability in terms of rainfall with Numerical Weather Prediction Systems (NWPSs) (Buzzi et al., 2013;

Fiori et al., 2014) and in terms of streamflow (Alfieri et al., 2012; Siccardi et al., 2005;

Silvestro and Rebora, 2014; Versini et al., 2014) even referring to hydrological nowcasting techniques (Borga et. al, 2011; Liechti et al., 2013; Silvestro et al., 2015a)

During the autumn 2011 two flash floods struck the Liguria Region of Italy causing a total of

19 victims and a large amount of damages. The first flash flood occurred the 25th October

2011; it affected the Cinque Terre coastal towns of Monterosso and Vernazza on the Eastern

Liguria Region and caused the flooding of Magra river.  The second event occurred 9 days later, the $4^{th}$ November, at about 50 km of distance and mainly affected the city of Genova with the flooding of Bisagno creek (see Figure 1).

Figure 1

They became two "school cases" studied by many scientists around the world during the last five years and they awaken the interest of the local authorities and of the civil protection actors regarding these type of calamities. Due to the large amount of damages and the numerous victims, they caused a general increase of the sensibleness of the citizens of the stricken areas regarding the natural hazards.

Both the events where characterized by very high rainfall intensities and a highly persistent localization. The V-shaped precipitation structure was observed in both cases, the rainfall cells were anisotropic with the dimension of major axis of 50-60 km oriented in the direction perpendicular to the coast and the dimension of minor axis of 5 to 15 km (see Rebora et al., 2013). The maximum hourly rainfall intensities measured by a gauge where around 160 mm during 4[th] November event and 150 mm during 25[th] October event, while the 24 hours accumulation where respectively around 500 and 540 mm. Figure 2 shows the maximum accumulated rainfall on 6 hours for the two events obtained merging the radar data from Italian national mosaic and gauge data with the algorithm described in Sinclair and Pegram (2005).

Figure 2

In both cases the effects in terms of discharge were important, the Bisagno creek (Area = 98 km$^2$) flooded the 4[th] of November reaching a peak flow with return period (T) around 30 yrs while Magra basin (Area = 1660 km$^2$) flooded the 25[th] of October reaching a peak flow with T around 50 yrs; in some small tributaries the peak flow had T larger than 100 yrs during both the events.

[revised manuscript text omitted]

Red lines represent the North West Italian regions.

[Figure]

Figure 2: Comparison of the 6 hours maximum accumulated rainfall (mm) for the events on 2011/10/25 (top panel) and on 2011/11/04 (bottom panel).

[Figure]

Figure 3: 2011/10/25, accumulated rainfall on 24 hours. Left panel, observed rainfall field; right panel, hypothetical rainfall field obtained by the rigid translation of the observed rainfall field from the original position to the Bisagno creek.

[Figure]

Figure 4: Schematization of the Flood Forecast Framework made by a downscaling model and a hydrological model. In this application the rainfall field is the one reported in figure

2.

[Figure]

Figure 5: An example of mixed-use curve definition. The green curve corresponds to the flood vulnerability function for the content of a 2-storey building, with mixed commercial and residential use: specifically retail trade at ground floor and residential at first floor. It is obtained by combining the 1-storey curve for generic residential (in red) with the 1-storey curve for retail trade (in blue). The yellow section of the graph represents the average height of the ground floor. The left part of the mixed curve is obtained by re-scaling the blue curve.

For higher values of water level, the function increases proportionally to the values that the residential curves assumes (red) in the left part of the graph.

It is assumed that the value of the ground floor is equal to 60% of the whole (2-storeys)

building.

[Figure]

Figure 6: Comparison between water depth – damage curves for content: retail trade (COM1)

building[blue], mixed retail trade (COM1) at first floor & RES at second floor[red], mixed retail trade (COM1) at first floor & residential (RES) at second and third floor[green].

[Figure]

[Figure]

Figure 7: Passerella Firpo reference section, Area: 93 km$^2$. Box plot of the peak flow generated by the FFF. On Y axis the peak flow is reported, on X axis the temporal aggregation scales (RSt) are reported. Diamonds represent the peak flow of the reference hydrograph. Each sub-panel shows results for a different spatial aggregation scale (RSs).

[Figure]

Figure 8. Streamflow scenarios derived by RSs=4 km and RSt=3 hrs. The hydrographs that lead to the peak flows with 10, 25, 50, 75, 90 percentiles were extracted.

[Figure]

Figure 9. Center of Genova city. Flood occurred on 9[th] October 2014. Comparison of the maximum flooding extent obtained through Telemac-2D and the field observations. The model was set in order to obtain the best fit between modeling and observations.

[Figure]

Figure 10: Perc10 scenario, inundation map and damage estimation. In blue scale the water level is reported. The damage is estimated at building scale in euro, the color scale ranges from low damage (green) to high damage (red).

[Figure]

Figure 11: Perc50 scenario, inundation map and damage estimation. In blue scale the water level is reported. The damage is estimated at building scale in euro, the color scale ranges from low damage (green) to high damage (red).

[Figure]

Figure 12: Perc90 scenario, inundation map and damage estimation. In blue scale the water level is reported. The damage is estimated at building scale in euro, the color scale ranges from low damage (green) to high damage (red).

[Figure]

Figure 13: Perc10 scenario, hazard level map compared with population potentially involved assigned to each building.

[Figure]

Figure 14: Perc50 scenario, hazard level map compared with population potentially involved assigned to each building.

[Figure]

Figure 15: Perc90 scenario, hazard level map compared with population potentially involved assigned to each building.